# Fatigue and Associated Factors in an Immune-Mediated Inflammatory Disease Population: A Cross-Sectional Study

**DOI:** 10.3390/jcm11092455

**Published:** 2022-04-27

**Authors:** Francesco Salvatore Iaquinta, Rosa Daniela Grembiale, Daniele Mauro, Ilenia Pantano, Saverio Naty, Cristina Cosco, Daniela Iacono, Emanuela Gaggiano, Annarita Ruggiero, Francesco Ciccia, Patrizia Doldo, Rocco Spagnuolo

**Affiliations:** 1Department of Health Sciences, Magna Græcia University of Catanzaro, 88100 Catanzaro, Italy; rdgrembiale@unicz.it (R.D.G.); saverio_naty@yahoo.it (S.N.); 2Department of Precision Medicine, L. Vanvitelli University of Campania, 80131 Naples, Italy; dranielmar@gmail.com (D.M.); ilenia.pantano@libero.it (I.P.); daniela.iacono@unicampania.it (D.I.); emanuela-gaggiano@libero.it (E.G.); annarita.ruggiero@alice.it (A.R.); francesco.ciccia@unicampania.it (F.C.); 3Department of Clinical and Experimental Medicine, Magna Græcia University of Catanzaro, 88100 Catanzaro, Italy; cosco.cristina@libero.it (C.C.); doldo@unicz.it (P.D.); spagnuolo@unicz.it (R.S.)

**Keywords:** immune-mediated inflammatory diseases, fatigue, PROMIS, inflammatory bowel disease, rheumatoid arthritis, spondyloarthritis

## Abstract

Fatigue is a main symptom of chronic diseases, including immune-mediated inflammatory diseases (IMIDs), such as inflammatory bowel disease (IBD) and inflammatory arthritis (IA); however, the pathophysiological mechanisms are not completely understood. The aim of this study was to assess the prevalence of fatigue and the associated factors in an IMIDs population. A control group, IBD, and IA patients, were enrolled. The PROMIS^®^ fatigue questionnaire was used to evaluate the symptoms. Information on demographics, anthropometrics, disease characteristics, and medications was collected for each participant. A total of 471 subjects (137 with IBD, 103 with IA, and 206 controls) were enrolled. IBD and IA patients reported greater fatigue than controls (*p* < 0.001, each). In univariate regression, patients with anxiety and depression were more likely to report fatigue (*p* = 1.40 × 10^−9^ and *p* = 3.80 × 10^−11^, respectively). Males, holding a high school diploma, and being employed were inversely correlated to the domain (*p* = 1.3 × 10^−5^; *p* = 0.003 and *p* = 0.005, respectively). The use of steroids and disease activity determined increased fatigue (*p* = 0.014 and *p* = 0.019; respectively). In the multivariate analysis, anxiety and depression remained associated (*p* = 0.002 and *p* = 1.3 × 10^−5^, respectively). IMIDs patients present increased fatigue compared with healthy subjects. Anxiety and depression are the main associated factors, suggesting a psychological component of the symptom; thus, a holistic management should be established.

## 1. Introduction

Fatigue is usually defined as “difficulty or inability to start or sustain activities.” In the healthy general population, reported fatigue ranges from 5% to 45% of the cases. Chronic fatigue, i.e., fatigue lasting more than 6 months, is experienced by 2–11% of individuals [1]. However, no common consensus has been reached on a definition of fatigue; therefore, it is difficult to determine the exact prevalence in both the general and diseased population.

Immune-mediated inflammatory diseases (IMIDs) are a group of chronic inflammatory conditions, including rheumatoid arthritis (RA), psoriatic arthritis (PsA), spondyloarthritis (SpA), and inflammatory bowel disease (IBD), that share the activation of common signaling pathways and the dysregulation of normal immune responses in genetically predisposed individuals [2]. Although clinical manifestations of IMIDs are extremely varied, fatigue represents one of the most common and disabling symptoms. As many as 80% of IBD patients with active disease and 50% of those in remission report fatigue, with significant social and economic costs [3]. In rheumatic diseases, the prevalence of fatigue varies from 40% to 80%, depending on the study and the measurement tool used [4].

Despite the significant burden of fatigue on a patient’s quality of life, its pathophysiological mechanisms are not fully understood. As a matter of fact, biological, physiological, and psychological mechanisms are likely to promote and maintain fatigue [5]. In inflammatory rheumatic diseases, fatigue is considered to be the result of many processes that can change over time. The inflammatory state has been associated with fatigue; however, many patients with rheumatic diseases experience fatigue, despite clinical and biological remission. Likely, inflammation could trigger the condition, but other mechanisms would be involved in maintaining it. In addition to inflammation, central nervous system alterations, neuroendocrine, sleep, and metabolic disorders have been suggested as factors; however, they cannot explain all cases. In fact, mood disorders such as depression and anxiety, pain, and psychosocial factors also contribute to the complex pathways that lead to chronic fatigue [6]. In IBD, the pathophysiology of fatigue is also considered to be multidimensional. Along with the inflammatory state, psychosocial comorbidities, nutritional deficiencies, metabolic alterations, lifestyle, and microbiota changes are all determinants of fatigue [3].

Among the components of fatigue, mood disorders are often confounding factors. Patients with IMID are at a higher risk of experiencing mood disorders than healthy controls [7]. Patients with rheumatoid arthritis and IBD have a higher prevalence of anxiety and depression, reporting similar results [8,9]. Furthermore, up to 40% of patients with SpA show depressive symptoms [10]. However, although depression has been strongly associated with fatigue in rheumatic diseases [11,12,13], it is unclear whether the two symptoms are linked by the same pathway, namely inflammation.

Patient perception and quality of life are generally assessed through patient-reported outcomes (PROs) that measure self-reported outcomes, including symptoms, disease perception, and health-related quality of life [14]. PROMs consist of a predetermined series of validated questions. However, accurately assessing and reporting fatigue is challenging. Due to its inherent subjectivity and the self-reported nature of the questionnaires used, the results can be significantly different. Furthermore, each questionnaire has strengths and weaknesses; in some cases, they lack measurement accuracy [15,16]. Lastly, fatigue can be considered a complex phenomenon that requires an adequate selection of PROMs, that is, a single element with respect to multiple elements, generic or disease specific, and multidimensional with respect to the overall scale score [17].

In this regard, the Patient-Reported Outcomes Measurement Information System (PROMIS^®^), is a set of widely validated and standardized item banks used to measure patient-reported outcomes (PROs) in many dimensions of health, such as mood disorders, pain, social life satisfaction, physical functioning, and fatigue [18], therefore useful for measuring fatigue as a single domain.

To date, no studies have evaluated fatigue in a cohort of IMIDs, including patients with IBD and inflammatory arthritis (IA). The main aim of this study was to assess the prevalence of fatigue in an IMIDs population, compared with a healthy control group, through the PROMIS^®^ questionnaire. As secondary aim, we intended to establish which variables could influence reported fatigue.

## 2. Materials and Methods

### 2.1. Study Population

Between March 2021 and August 2021, consecutive outpatients attending the Rheumatology and IBD Departments of the Magna Græcia University of Catanzaro and Luigi Vanvitelli, the University of Naples, were recruited into the study. A control population was also enrolled. While waiting for the scheduled visit at the clinical department, all patients were invited to complete questionnaires on fatigue, anxiety, and depression using PROMIS^®^ items. Subjects were instructed on how to fill in the questions, and then they answered without any help from the healthcare professionals. Items were scored as detailed below.

Participants ≥18 years of age and able to provide informed consent were included, while the following have been set as exclusion criteria: non-Italian speaking subjects; those unable to read or understand the questions; individuals with cognitive impairment, and patients with a confirmed diagnosis and/or in treatment for mood disorders. Patients had a confirmed diagnosis of Crohn’s Disease (CD) and Ulcerative Colitis (UC) based on endoscopic, histological, and clinical criteria; or PsA, SpA, and RA diagnosed according to CASPAR [19], ASAS [20], and EULAR/ACR [21] classification criteria, respectively. Demographic and anthropometric characteristics were acquired for all subjects.

Patients underwent a comprehensive assessment of disease characteristics, including disease duration and disease activity assessed through ASDAS (CRP) for axial spondyloarthritis (with scores ≥1.3 identifying the disease as active) [22]; DAPSA28 (CRP) evaluated psoriatic arthritis (a value >4 defined active disease) [23]; DAS28 (CRP), was used for rheumatoid arthritis (with scores >2.6 defined as active disease) [24]; the Harvey Bradshaw Index (HBI) [25] assessed Crohn’s disease (a score >7 indicated active disease), and finally, the Mayo Score (MS) was used for ulcerative colitis (a score >5 was used to identify active disease) [26]. C-reactive protein (CRP), erythrocyte sedimentation rate (ESR), and information on treatment such as the use of steroids, disease-modifying antirheumatic drugs (DMARDs), and mesalamine was also collected.

### 2.2. Ethics Approval

The ethics committee of Magna Græcia University approved this study on 17 September 2020 (Protocol No. 340), and it was conducted following the Declaration of Helsinki principles. Each participant provided written informed consent.

### 2.3. Instruments

Fatigue was evaluated by the Italian version (Short Form 8a, v1.0) of the Patient-Reported Outcome Measurement Instrument System (PROMIS^®^). The PROMIS^®^ Fatigue item bank uses five-point Likert scales ranging from 1 (Never/Not at all) to 5 (Very much/Always). The fatigue survey explored symptoms and their interference with daily activity. The anxiety (Short Form 8a, v1.0) and depression (Short Form 8a, v1.0) domains were also rated by PROMIS^®^ with a scale ranging from 1 (Never) to 5 (Always). All domains looked at the condition for the previous seven days.

PROMIS^®^ measures use a *t*-score metric with a mean of 50 and a standard deviation of 10 from the general US population. A cut-off *t*-score of 50 was used to dichotomize the scores. More precisely, *t*-scores ≥ 50 suggest the presence of the condition and *t*-scores < 50 indicate its absence.

### 2.4. Data Analysis

Continuous variables are expressed as mean ± standard deviation (SD) if normally distributed and as median with interquartile range (IQR) otherwise. Categorical data are shown as frequencies and percentages. Statistical analysis was conducted to evaluate differences among the groups in the demographic and anthropometric characteristics, as well as the domains studied. The Kruskal–Wallis, with post-hoc pairwise test (Bonferroni correction), the Mann–Whitney U test, or the Chi-squared test were used to perform the analysis between groups and within groups, as appropriate. In addition, univariate binary logistic regression was performed in the IMIDs group to determine the variables associated with fatigue. Then, statistically significant variables in the univariate analysis were included in a multivariable logistic regression model, adjusted for age, gender, body mass index (BMI), and disease to define independent risk factors. Logistic regression results are presented as odds ratio (OR) and 95% confidence interval (95% CI). Analysis was performed using IBM SPSS^®^ Statistics (IBM, Armonk, NY, USA), Version 26; *p* values < 0.05 were considered statistically significant.

## 3. Results

### 3.1. Study Population 

The demographic, anthropometric, and clinical characteristics of the participants are presented in Table 1. Briefly, 471 subjects were enrolled: 137 (30.7%) with IBD (*n* = 39, (29%) Crohn’s disease and *n* = 98, (71%) ulcerative colitis); 103 (23.1%) with inflammatory arthritis (*n* = 31, (30%) rheumatoid arthritis and *n* = 72, (70%) spondyloarthritis), and 206 (46.2%) controls. Age and BMI did not differ between groups. 

For demographic characteristics, we found a significant overall difference in gender across all groups (*p* = 7.1 × 10^−5^) with specific differences between controls and IA (*p* = 1.09 × 10^−4)^, IBD, and IA (*p* = 8.3 × 10^−5^). Educational level differed between controls and IBD (*p* = 0.016), IBD, and IA (*p* = 0.022), with an overall difference among groups (*p* = 0.027). Finally, IBD patients smoked less than controls (*p* = 1.2 × 10^−5^) and IA patients (*p* = 0.004), with a global difference across all groups (*p* = 6.5 × 10^−5^). Regarding mood disorders, a statistically significant difference was found across all groups in number of subjects experiencing both anxiety (*p* = 0.002) and depression (*p* = 0.004), and in the *t*-scores (*p* = 1.34 × 10^−4^ and *p* = 4.46 × 10^−4^, for anxiety and depression, respectively). In more detail, IMIDs patients experienced anxiety more frequently than controls (*p* = 0.001 and *p* < 0.016, for IBD and IA patients, respectively), with higher *t*-scores in IBD vs. controls (*p* = 1.68 × 10^−6^) and IA vs. controls (*p* = 0.024). Lastly, IBD and IA patients reported depression more than healthy subjects (*p* = 0.034 and *p* = 0.001, respectively), but with regards to depression *t*-score, only IA patients had greater symptoms than controls (*p* = 3.87 × 10^−4^). No other differences were found between groups.

### 3.2. IMIDs Patients Have Increased Fatigue Compared with Healthy Controls

Fatigue was reported by 71 (34.5%) control subjects, with a median *t*-score of 48 (43–52). A total of 75 (54.7%) IBD patients expressed fatigue, with a median *t*-score of 52 (46–60) and 73 (70.9%) of IA patients were fatigued, reporting a median *t*-score of 54 (49–63). Figure 1 shows comparisons between groups for the PROMIS^®^ fatigue item. Both IBD and IA patients experienced significantly increased fatigue compared with controls (*p* < 0.001, each), but no differences were found between the IMIDs group.

### 3.3. Variables Associated with Fatigue

As shown in Table 2, in the univariate regression model, males presented minor fatigue (OR 0.29, 95% CI (0.17–0.51); *p* = 1.3 × 10^−5^). Among demographic characteristics, holding a high school diploma and having an occupation were inversely correlated to the domain (OR 0.41, 95% CI (0.23–0.74); *p* = 0.003 and OR 0.47, 95% CI (0.28–0.80); *p* = 0.005, respectively). Moreover, steroid treatment and disease activity determined increased fatigue scores (OR 3.20, 95% CI (1.27–8.08); *p* = 0.014 and OR 2.12, 95% CI (1.13–3.97); *p* = 0.019; respectively). As expected, patients with anxiety and depression were more than six times and nine times more likely to show fatigue (OR 5.90, 95% CI (3.32–10.49); *p* = 1.40 × 10^−9^ and OR 9.70, 95% CI (4.95–19.03); *p* = 3.80 × 10^−11^, respectively). Finally, IA patients had two-fold the risk of reporting fatigue (OR 2.01, 95% CI (1.17–3.46); *p* = 0.012). In a univariate logistic regression model for IBD only, the results were virtually the same, except for disease activity, which was not an associated factor (see Appendix A). Conversely, in the IA group alone, patients with active disease were more likely to show fatigue (see Appendix A).

### 3.4. Mood Disorders Are Strongly Associated with Fatigue in IMIDs Patients

Table 3 shows multivariable logistic regression results. In the model, anxiety and depression remained independently associated (OR 6.15, 95% CI (1.51–6.59); *p* = 0.002 and OR 5.92, 95% CI (2.66–13.17); *p* = 1.3 × 10^−5^, respectively). In separate multivariate models for disease, mood disorders and steroids were linked to fatigue in IBD patients (see Appendix A), whereas active disease but not anxiety was associated with fatigue in IA patients (see Appendix A). 

## 4. Discussion

Patients with immune-mediated inflammatory diseases are at higher risk of reporting impaired quality of life than the general population [27,28,29,30]. Fatigue is one of the many symptoms that patients with IBD and IA experience throughout their lifetimes. The prevalence of fatigue in IMIDs patients varies according to demographic, sociocultural, and geographic characteristics, as well as the measurement tool used in the study. However, overall, as other authors have shown, more than 55% of IBD patients suffer from fatigue [31], while 61% of the IA population is fatigued [32].

Our results confirm previous findings, suggesting that the symptom represents a significant burden on patients; indeed, in our study, IMIDs patients had increased fatigue compared with a healthy population, but scores did not differ between patient groups. We also analyzed the prevalence of anxiety and depression that are commonly associated with fatigue, and most patients with IMID experienced greater anxiety, compared to healthy subjects (*p* = 1.68 × 10^−6^ for IBD and *p* < 0.024 for IA patients, respectively). As for depressive symptoms, only patients with IA had higher scores than controls (*p* = 3.87 × 10^−4^).

To date, this is the first study in which PROMIS^®^ fatigue, anxiety, and depression item banks have been used simultaneously to assess the symptoms and compare an IMIDs population with healthy controls. Mancina et al. [33] showed that among physical and psychological symptoms, 60% of IBD patients reported to be fatigue closely associated with gastrointestinal symptoms, such as nausea and vomiting, belly pain, and diarrhea, further suggesting its multidimensionality. Furthermore, a longitudinal analysis of quality of life in IBD patients through PROMIS^®^ domains, showed a worsening of fatigue scores with worsening disease activity [34]. On the other hand, the shortened form of the PROMIS^®^ fatigue survey has been validated in patients with rheumatoid arthritis [35] and spondyloarthritis [36].

In the present study, the univariate logistic regression showed that several variables were associated with reported fatigue. Specifically, males, having a high school diploma, and being employed were protective factors for fatigue, while illness, disease activity, steroids treatment, and anxiety and depression symptoms increased the risk. However, in multivariable logistic regression, adjusted for age, BMI, gender, and disease, only anxiety and depression remained as independent related factors. In a cross-sectional study, López-Medina et al. [37], studying a cohort of 2251 SpA patients, reported that female gender and emotional status were independently associated with fatigue. Other studies conducted in different IA populations demonstrated that depression and sleep disturbances increased the odds of fatigue [38,39,40]. Similar results were observed for IBD patients; in the study of Chavarría el al. [41], in addition to anxiety and depression, fatigue was also associated with steroid treatment.

In our study, steroids were only associated in the univariate analysis. Furthermore, Schreiner et al. [31] reported that active disease was an associated factor of fatigue as confirmed in our univariate analysis. Active disease also remained associated, both in the univariate and the multivariate analysis, for IA patients alone.

These findings suggest that while mood disorders imply a higher risk of experiencing fatigue independently of the disease, active disease, that is, the inflammatory state, is an additional element determining fatigue, especially in inflammatory arthritis. Therefore, it is likely that even within IMIDs, fatigue may be caused by different mechanisms.

The main strength of this study is the use of specific questionnaires focused on single domains that can better detect symptoms. Indeed, PROMIS^®^ measures have demonstrated good clinical validity in a range of chronic conditions [42,43]. A recent psychometric evaluation of the Italian custom 4-item Short Form of the PROMIS^®^ anxiety survey, performed by Liuzza et al. [44], showed acceptable reliability, particularly for individuals with higher levels of anxiety, such as IMIDs patients. Therefore, the data obtained in our study confirm the applicability of PROMIS^®^ measurement instruments in this context. Indeed, it should be notice that as a symptom, fatigue can only be assessed by self-report, and although it is possible, under certain circumstances, to quantify performance fatigability (resulting from the contractile activity of muscles, and the ability of the nervous system to provide a suitable activation signal) through quantifiable measures, it does not include perceived fatigability and thus, does not offer a reliable estimate of the symptom [45].

However, this study has some limitations. First, the data were collected in only two different centers, so it still remains uncertain whether the results can be generalized, even in a different population. Furthermore, although univariate and multivariable logistic regression showed associated factors, a larger sample size is needed to further detect demographic and clinical characteristics associated with fatigue. Second, the cross-sectional design of the study does not have the ability to track potential changes in reported symptoms over time. Indeed, it is essential to clarify whether and the extent to which specific treatment, such as biological drugs, steroids, or cognitive behavioral therapy, could improve health-related quality of life and modify experienced fatigue accordingly. On the other hand, we did not assess the presence of diabetes, hypothyroidism, anemia, and other diseases that can cause fatigue; thus, further comprehensive studies are necessary. Given the strength of the association between mood disorders and fatigue in the multivariate analysis, it is conceivable that fatigue could be considered a surrogate marker for anxiety and depression; indeed, it is deemed a main determinant of depression, resulting in the need of an in-depth investigation of subclinical mood disorders among IMIDs patients. Thus, it cannot be excluded that the treatment of underdiagnosed conditions could lead to an improvement in fatigue. However, since a specific diagnosis for these conditions requires a thorough evaluation by a psychiatrist, fatigue on its own could suggest the presence of an underlying disorder, especially in patients with the worst scores, but this needs to be integrated in the complex interplay of different pathogenic mechanisms occurring in inflammatory diseases, as reported in the multivariate analysis of fatigue in IA patients (Appendix A), where both demographic and clinical features were associated with the symptom.

## 5. Conclusions

IMIDs patients present increased fatigue compared with healthy subjects. Among the different variables, anxiety and depression are independently associated factors, suggesting a psychological component. However, sociodemographic and disease-related elements cannot be excluded. Therefore, more studies are required to better define this unmet need. Although the exact pathogenesis is still unclear, fatigue is a disabling symptom in inflammatory diseases; therefore, researchers must focus their efforts on achieving a common accepted definition and on improving measurement accuracy. On the other hand, health care practitioners must investigate its presence, the impact on a patient’s quality of life, and the underlying components to establish a correct holistic and personalized management.

## Figures and Tables

**Figure 1 jcm-11-02455-f001:**
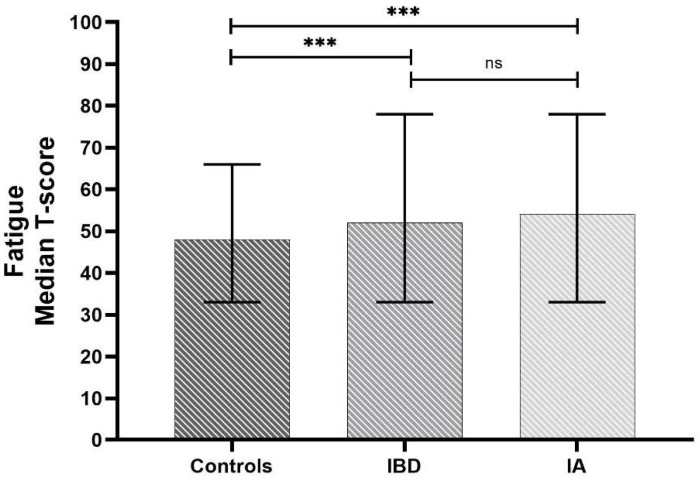
Fatigue in IMIDs patients (IBD: Inflammatory Bowel Disease, IA: Inflammatory Arthritis) as measured by PROMIS^®^ scales. Data are presented as median and range. The *p*-values were calculated by the Kruskal-Wallis non-parametric test for independent samples, Bonferroni correction. (*** *p* < 0.001; ns: non-significant).

**Table 1 jcm-11-02455-t001:** Characteristics of the study population.

	Healthy Controls (*n* = 206)	IBD (*n* = 137)	IA (*n* = 103)	*p*
**Demographic and** **Anthropometric**				
Age (years)	53 (37–65)	49 (39–58)	53 (42–61)	0.157
Gender, *n* male (%)	126 (61.2)	87 (63.5)	39 (37.9)	**7.1 × 10^−5^** ^a^
BMI (Kg/m^2^)	25 (22–27)	24 (22–27)	25 (23–27)	0.165
Smoking, *n* = yes (%)	91 (44.2)	29 (21.2)	39 (37.9)	**6.5 × 10^−5^** ^b^
Physical Activity, *n* = yes (%)	73 (34.5)	56 (40.9)	27 (26.2)	0.061
High school diploma, *n* = yes (%)	121 (58.7)	98 (71.5)	59 (57.3)	**0.027** ^c^
Marital status, *n* = not married (%)	73 (35.4)	34 (24.8)	26 (25.2)	0.056
Occupation, *n* = yes (%)	84 (40.8)	69 (50.4)	47 (45.6)	0.213
**Disease characteristics**				
Crohn’s Disease	-	39 (28.5)	-	-
Ulcerative Colitis	-	98 (71.5)	-	-
Rheumatoid Arthritis	-	-	31 (30.1)	-
Spondyloarthritis	-	-	72 (69.9)	-
Disease duration (years)	-	12 (7–18)	10 (5–14)	**0.012**
Active Disease, *n* = yes (%)	-	31 (22.6)	34 (33.0)	0.073
ESR (mm/h)	-	8 (4–16)	10 (5–19)	**0.021**
CRP (mg/dL)	-	0.3 (0.3–0.6)	0.4 (0.2–1.5)	0.651
**Medications n (%)**				
Steroids	-	18 (13.1)	15 (14.6)	0.751
Biological DMARDs	-	26 (19.0)	64 (62.1)	**8.16 × 10^−16^**
Methotrexate	-	1 (0.7)	40 (38.8)	**8.27 × 10^−15^**
Mesalamine	-	122 (89.1)	-	-
**Mood disorders**				
Anxiety, *n* = yes (%)	96 (46.6)	88 (64.2)	63 (61.2)	**0.002** ^d^
Anxiety, *t*-score	49 (43–55)	55 (46–63)	53 (46–60)	**1.34 × 10^−4^** ^e^
Depression, *n* = yes (%)	60 (29.1)	55 (40.1)	49 (47.6)	**0.004** ^f^
Depression, *t*-score	45 (38–51)	45 (38–56)	49 (38–56)	**4.46 × 10^−4^** ^g^

*p*-values for medications and disease characteristics were obtained excluding healthy controls. Significant *p*-values across all groups are displayed in bold. Post-hoc analysis: ^a^: Controls vs. IA: *p* = 1.09 × 10^−4^; IBD vs. IA: *p* = 8.3 × 10^−5^; ^b^: Controls vs. IBD: *p* = 1.2 × 10^−5^; IBD vs. IA: *p* = 0.004; ^c^: Controls vs. IBD: *p* = 0.016; IBD vs. IA: *p* = 0.022; ^d^: Controls vs. IBD: *p* = 0.001; Controls vs. IA: *p* = 0.016; ^e^: Controls vs. IBD: *p* = 1.68 × 10^−6^; Controls vs. IA: *p* = 0.024; ^f^: Controls vs. IBD: *p* = 0.034; Controls vs. IA: *p* = 0.001; ^g^: Controls vs. IA: *p* = 3.87 × 10^−4^. Abbreviations: IBD: inflammatory bowel disease; IA: inflammatory arthritis; BMI: body mass index; ESR: erythrocyte sedimentation rate; CRP: C-reactive protein; Biological DMARDs: biological disease-modifying antirheumatic drugs.

**Table 2 jcm-11-02455-t002:** Univariate logistic regression model for fatigue in IMIDs.

Variables	OR (CI 95%)	*p*
Age	1.02 (1.00–1.04)	0.108
BMI	1.09 (0.99–1.19)	0.072
Gender		**1.3 × 10^−5^**
Male	0.29 (0.17–0.51)	
Female	1	
Disease		**0.012**
IA	2.01 (1.17–3.46)	
IBD	1	
Smoking		0.543
Yes	1.20 (0.67–2.15)	
No	1	
High school diploma		** 0.003 **
Yes	0.41 (0.23–0.74)	
No	1	
Physical activity		0.244
Yes	0.72 (0.42–1.25)	
No	1	
Marital status, single		1
Yes	1.00 (0.55–1.82)	
No	1	
Occupation		** 0.005 **
Yes	0.47 (0.28–0.80)	
No	1	
Disease duration	1.01 (0.98–1.05)	0.449
Disease activity		** 0.019 **
Yes	2.12 (1.13–3.97)	
No	1	
ESR	1.02 (1.00–1.04)	0.109
CRP	1.02 (0.89–1.16)	0.830
Steroids treatment		** 0.014 **
Yes	3.20 (1.27–8.08)	
No	1	
Biological treatment		0.132
Yes	1.52 (0.88–2.64)	
No	1	
Anxiety		**1.40 × 10^−9^**
Yes	5.90 (3.32–10.49)	
No	1	
Depression		**3.80 × 10^−11^**
Yes	9.70 (4.95–19.03)	
No	1	

IMIDs: immune-mediated inflammatory diseases; OR: odds ratio; CI: confidence interval; BMI: body mass index; IA: inflammatory arthritis; IBD: inflammatory bowel disease; ESR: erythrocyte sedimentation rate; CRP: C-reactive protein.

**Table 3 jcm-11-02455-t003:** Multivariable logistic regression model for fatigue in IMIDs.

Variables	OR (CI 95%)	*p*
Age	0.98 (0.96–1.01)	0.224
BMI	1.14 (1.00–1.29)	0.050
Gender		0.092
Male	0.53 (0.25–1.11)	
Female	1	
Type of disease		0.085
IA	1.89 (0.92–3.84)	
IBD	1	
High school diploma		0.143
Yes	0.51 (0.21–1.26)	
No	1	
Occupation		0.108
Yes	0.54 (0.25–1.15)	
No	1	
Disease activity		0.121
Yes	1.89 (0.85–4.23)	
No	1	
Steroids treatment		0.054
Yes	3.06 (0.98–9.55)	
No	1	
Anxiety		**0.002**
Yes	6.15 (1.51–6.59)	
No	1	
Depression		**1.3 × 10^−5^**
Yes	5.92 (2.66–13.17)	
No	1	

IMIDs: immune-mediated inflammatory diseases; OR: odds ratio; CI: confidence interval; BMI: body mass index; IA: inflammatory arthritis; IBD: inflammatory bowel disease.

## Data Availability

The data presented in this study are available on request from the corresponding author. The data are not publicly available due to personal information in the database.

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
