# Peer review of "Fatigue and Associated Factors in an Immune-Mediated Inflammatory Disease Population: A Cross-Sectional Study"

_jcm, 2022, doi:10.3390/jcm11092455_

Round 1

Reviewer 1 Report

In this study, the authors utilise their access to the ROMIS® fatigue, anxiety and depression banks to examine fatigue as a complication of disease and its correlation with disease characteristics in uni-, and multivariate analysis. The study appears to be suitably designed and the results, whilst not ground breaking are logical, and the conclusions appear balanced. Ultimately, the multivariate analysis throws up depression and anxiety as independent variables contributing to fatigue. There are some corrections and consideration I would recommend.

Major

Results

  • Better descriptive statistics would help with patient characteristics. It is not clear for example if statistics comparing gender differences are significant across all groups or just between IBD and IA for example. Could more effective gender matching have helped in this study set up? Basically, a clearer statistical comparison of characteristics between groups would be helpful here.
  • Given the strength of anxiety and depression correlations across uni, and multi-variate analysis with fatigue, the implication becomes one of fatigue being a surrogate marker for these symptoms, and whether the patients with worst fatigue in the IA and IBD cohorts have a subclinical depression. Could this have been assessed, and is greater examination of this aspect required? At the very least, this should be discussed in greater detail

Minor

  • Is there any scope for utilising non patient reported quantifiable measures of fatigue in the future, Using either EEG and cognitive information processing from ERP? I imagine not, but could be discussed briefly

Reviewer 2 Report

In this study, the authors examine the relationship between inflammatory disease and fatigue. They test this using a questionnaire-based experimental model where patients with inflammatory bowel disease (IBD) and inflammatory arthritis (IA) were asked to score themselves on the PROMIS fatigue scale.

It was unclear how the questionnaire was administered -- whether patients completed the questionnaire in a clinic, in the presence of a healthcare professional or whether they were given the questionnaire to take home and complete on their own time.

Were any of the patients that reported depression on any medications specifically to treat depression?

The error bars in Figure 1 are difficult to read and should be changed to a different color (black?).

Round 2

Reviewer 1 Report

All comments addressed